# Implementation fidelity of a community-based *Aedes aegypti* breeding site elimination program for dengue control in southeastern Mexico

Miguel Mazariego-Longoria[1], Carlos Rojas-Arbeláez[1]*, Héctor Vélez-Santamaría[2],
Abel Jiménez-Alejo[3], Ariane Dor[4], Guillermo Rúa-Uribe[5]

1  Grupo Epidemiología, Facultad Nacional de Salud Pública, Universidad de Antioquia, Medellín, Antioquia, Colombia, 2  Grupo de estudios en Pedagogía, Infancia y Desarrollo Humano, Facultad de Educación, Universidad de Antioquia, Medellín, Antioquia, Colombia, 3  Centro de Estudios Tecnológicos Industrial y de Servicios, núm, 116 "Antonia Nava de Catalán, Acapulco, Guerrero, México, 4  Secretaría de Ciencia, Humanidades, Tecnología e Innovación (SECIHTI), comisionada en El Colegio de la Frontera Sur – Unidad Tapachula, Tapachula, Chiapas, México, 5  Facultad de Medicina, Grupo Entomología Médica, Universidad de Antioquia, Medellín, Antioquia, Colombia

* carlos.rojas@udea.edu.co

## Abstract

### Introduction

Dengue is a globally significant arboviral disease in tropical regions such as southeastern Mexico. The "Healthy Environments and Communities" (E&CS) program, led by the Health Secretariat, promotes community practices to eliminate *Aedes aegypti* breeding sites. However, factors influencing its implementation and fidelity remain unclear.

### Objective

To determine the factors affecting fidelity in implementing the E&CS program targeting breeding site elimination in Río Florido, Tapachula.

### Methods

An observational mixed-methods study was conducted. The quantitative component evaluated fidelity among health personnel using a CFIF-based instrument, measuring Content Details, Frequency, Duration, and Coverage. Independent-samples t-tests compared mean fidelity scores between subgroups (DSVII and CSLL). The qualitative component included semi-structured interviews with health personnel and community members, coded using a priori CFIR categories and emergent themes to identify barriers and facilitators. Frequencies of barriers and facilitators among health personnel subgroups were compared using Fisher's exact test, and chi-square tests assessed differences in distribution.

**Data availability statement:** All survey and interviews files are available from Universidad de Antioquia, open data repository at https://datosdeinvestigacion.udea.edu.co/dataverse/fac-salud-publica.

**Funding:** Funded by TDR, the Special Programme for Research and Training in Tropical Diseases, hosted at the World Health Organization and co-sponsored by UNICEF, UNDP, the World Bank, and WHO (TDR grant number B40323). http://www.who.int/tdr/about/en/. The funders had no role in study design, data collection and analysis, decision to publish, or preparation of the manuscript.

**Competing interests:** The authors have declared that no competing interests exist.

## Results

Overall mean fidelity was 91.2% (95% CI: 88.5–93.4%). By construct, Content Details reached 92.4%, Frequency 89.8%, Duration 90.5%, and Coverage 91.3%. Among health personnel (n = 23; 142 responses), Fisher's exact test showed no significant differences (p > 0.05). After combining responses, the distribution did not differ from chance ($\chi^2 = 9.73$, df = 6, p = 0.136). In the Río Florido community, significant differences were observed ($\chi^2 = 83.16$, df = 11, p < 0.001). Main barriers included "insufficient attention from the health center" and perceiving the Health Secretariat as responsible for dengue. Key facilitators were recognition of E&CS, respectful relationships with health personnel, and belief in program success.

## Conclusion

High fidelity of the E&CS program (>90%) in Río Florido was accompanied by a differentiated pattern of barriers and facilitators, balanced among health personnel but uneven in the community. Addressing these factors through an implementation research (IR) approach could strengthen the sustainability and effectiveness of dengue control strategies.

## 1. Introduction

Dengue is one of the most globally significant arboviral diseases, with a concerning increase in incidence over recent decades, particularly in tropical regions such as Latin America and the Caribbean [1]. In Mexico, dengue represents a persistent public health threat, with substantial epidemiological and social impacts, particularly in southeastern states such as Chiapas, which borders Guatemala [2]. The southern border region of Mexico is recognized as an endemic area for dengue, consistently experiencing recurrent outbreaks and epidemics that have significantly affected local health conditions and the regional economy [2]. However, dengue transmission is not limited to this area; other southern states, including Guerrero and Veracruz, also report recurrent viral circulation and the presence of all four dengue serotypes, indicating a broader regional pattern already recognized in Mexico [2].

The Chiapas State Health Secretariat (SSCH) reported that, as of epidemiological week (EW) 33 in 2025, 4,083 probable dengue cases, 430 confirmed cases, two deaths, and a case fatality rate of 0.72% had been recorded [3]. In comparison, during the same week in 2024, 2,490 confirmed cases and six deaths were reported, indicating a declining trend in dengue incidence in Chiapas in 2025 [3]. Consistently, the 2025 Epidemiological Overview from the Mexican Health Secretariat (SSA) for EW 33 also documents a notable decrease in confirmed cases compared to the same period in 2024 [3].

Importantly, entomological evidence from rural localities in southern Chiapas, including Río Florido, shows that *Aedes aegypti* and *Aedes albopictus* coexist in the area, with marked seasonal fluctuations and persistent egg abundance throughout

the year. Baseline data from longitudinal oviposition monitoring in Río Florido demonstrate higher *Aedes* activity during the rainy season and stable presence during the dry season, reflecting continuous vector pressure in the community [4]. These ecological conditions highlight the relevance of targeting breeding-site elimination and community engagement strategies, as the local entomological context may influence both the implementation requirements and the fidelity of programs such as E&CS.

Dengue prevention in Mexico relies on vector control, with a particular focus on eliminating *Aedes aegypti* breeding sites, the primary vector of the disease. This mosquito lays eggs in a variety of artificial containers with clean water inside and around homes, most commonly flowerpots, buckets, barrels, and uncovered or poorly covered water tanks, which represent the most productive breeding sites [5]. Therefore, eliminating these mosquito breeding sites is a fundamental step in dengue control.

Since 2001, the Mexican Health Secretariat (SSA) has implemented the Healthy Environments and Communities (E&CS, for its Spanish acronym) program, an institutionally driven intervention that aims to foster social participation in health promotion and disease prevention through the transformation of physical and social environments. Although it is a community-based intervention, its execution is coordinated by Health Secretariat personnel nationwide. The general objective of this intervention is strengthening community and municipal engagement in health promotion to create healthy, safe, and hygienic environments, thereby reducing vector presence and the risk of arboviral transmission.

In the context of *Ae. aegypti* control, the E&CS program promotes actions aimed at eliminating vector breeding sites through active community involvement. Its specific objectives are: 1) participatory diagnosis of health risks and environmental hazards; 2) joint identification of intervention priorities; 3) development of a community action plan; 4) execution of activities such as cleanup campaigns, home visits, and educational workshops; and 5) monitoring and evaluation of results with feedback to the community. These stages are facilitated by health personnel in collaboration with community leaders, health committees, and other social actors, with the goal of fostering sustainable practices that reduce vector presence and improve environmental conditions [6].

The effectiveness of such interventions depends not only on their design but also on how they are implemented in practice. Implementation fidelity, the degree to which an intervention is delivered as intended, is a critical component to ensure the expected impact [7]. Previous studies have reported that barriers such as lack of resources, low community participation, or poor execution can significantly reduce the effectiveness of community-based interventions [8,9].

Implementation Research (IR) provides conceptual and methodological tools to analyze how health interventions, programs, and policies are delivered in real-world settings [10]. One of its strengths is the use of multiple theories, models, and conceptual frameworks [11], which guide the study of intervention implementation across different phases and dimensions.

In this study, as main objective, we evaluated the implementation fidelity of the E&CS program in the rural community of Río Florido, Tapachula, Chiapas. We analyzed both the fidelity of Health Secretariat personnel to key program components and the perceived barriers and facilitators from institutional and community perspectives.

## 2. Materials and methods

### 2.1. Study site

The study was conducted in Río Florido, a communal land (ejido) in the southwestern part of Tapachula Municipality, Chiapas, Mexico (14°51'17.140 N, 92°20'27.166 W; Fig 1). The site is situated at 59 meters above sea level and has a predominant tropical monsoon climate ("Am"), with an average annual temperature of 26.5°C and 2,700 mm of annual precipitation. Río Florido covers an area of 20 hectares, with 887 inhabitants, 228 households, and an average of 4.59 inhabitants per household. Agriculture is the main economic activity in Río Florido [12].

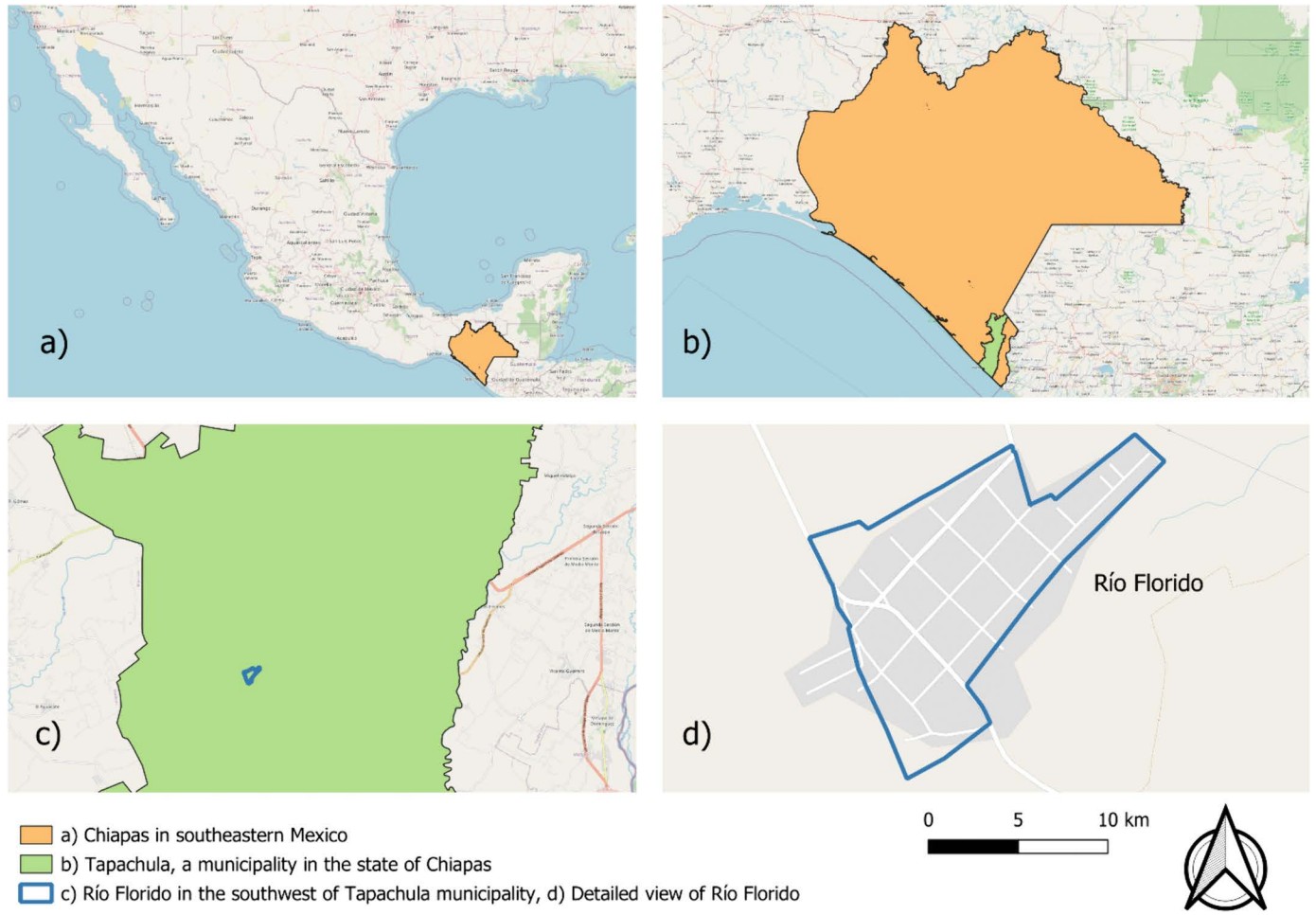

**Fig 1. Location of Río Florido in Chiapas, Mexico.** a) Chiapas in southeastern Mexico. b) Tapachula, a municipality in the state of Chiapas. c) Río Florido in the southwest of Tapachula municipality d) Detailed view of Río Florido. Source: Map created by the author using QGIS 3.40.7 with open-access geographical data from INEGI, licensed under CC BY 4.0. [11].

## 2.2. Study design

A mixed-methods observational study was conducted, integrating qualitative and quantitative techniques to evaluate the implementation fidelity of the E&CS program in Río Florido. This approach allowed for a comprehensive understanding of the implementation processes from the perspectives of health personnel and the community.

## 2.3. Conceptual frameworks

The analysis was guided by two widely used conceptual frameworks in Implementation Research: The Consolidated Framework for Implementation Research (CFIR) [13], which systematically examines internal and external contextual factors affecting implementation (qualitative data), and the Conceptual Framework for Implementation Fidelity (CFIF) [7], which considers dimensions such as adherence to Content Details, Duration, Frequency, and Coverage (quantitative data).

## 2.4. Pilot testing

A pilot study was conducted to pre-evaluate the data collection instruments under real field conditions and to verify whether the selected variables were adequate in terms of clarity, application time, and participant comprehensibility. Semi-structured interviews (qualitative) and structured questionnaires (quantitative) were administered to health personnel (n = 4) from the Health Promotion area of the health center located in Segunda Sección de Tinajas, municipality of Tapachula, Chiapas. Segunda Sección de Tinajas is a rural locality situated approximately between 36 and 44 meters above sea level and is part of the coastal plain of the Soconusco region; it has an estimated population of 757 inhabitants and high coverage of basic services such as piped water, electricity, and sanitation [14].

Semi-structured interviews were also conducted with residents of the ejido Francisco I. Madero (n = 10), selected due to its sociodemographic similarity to Río Florido, the main study site. This ejido is located in the municipality of Tapachula at approximately 29 meters above sea level and has an estimated population of 678 inhabitants; the available basic housing and service indicators describe demographic and dwelling characteristics typical of rural communities in the coastal plain of the Soconusco region [15].

## 2.5. Population and sampling

For the qualitative approach, convenience sampling was used with all health personnel linked to the E&CS program in Tapachula, including 23 workers from two components of the Health Secretariat: The Health Promotion area of Health District VII (DSVII, n = 18) and the Llano de la Lima Health Center (CSLL, n = 5). Proportional stratified sampling was used to select residents of Río Florido. Based on an updated household count conducted by the principal investigator, the total population in the urban area was estimated at 233 individuals. A sample size of 80 was calculated, with an additional 20% added to compensate for potential non-responses, resulting in 96 participants (41.2% of the estimated population). Semi-structured interviews explored perceptions of E&CS program implementation. The community was divided into four zones to facilitate systematic household selection.

For the quantitative approach, convenience sampling was applied to the same health personnel carrying out E&CS activities in the two SSA components (DSVII and CSLL). Residents of Río Florido were not included, as implementation fidelity was assessed only for health personnel.

In both approaches, inclusion and exclusion criteria were applied. For health personnel, inclusion criteria were direct participation in E&CS dengue prevention activities and at least six months of experience in the program. Exclusion criteria were refusal to provide informed consent, assignment to areas unrelated to E&CS, and less than six months of experience. For Río Florido residents, inclusion criteria were adults available to participate in interviews. Exclusion criteria included refusal to consent, minors, individuals under the influence of drugs or alcohol, those with disabilities or unable to respond to interviews, and temporary residents or visitors.

The recruitment period for study participants began on 16/12/2022 and ended on 06/03/2023.

## 2.6. Qualitative data collection

Semi-structured interviews were conducted with health personnel (n = 23) and community members (n = 96). Two interview guides were developed based on the CFIR, one for health personnel and one for residents of Río Florido (see supplementary material). Two CFIR domains were selected to guide data collection and analysis: intervention characteristics and characteristics of the individuals involved.

For health personnel, constructs examined included complexity, design and quality of delivery, and strength and quality of evidence, providing insights into perceived difficulty, available resources, and institutional support. For Río Florido residents, beliefs and knowledge about the intervention were explored, including community perceptions, treatment by health personnel, and local dengue prevention knowledge. Distinguishing these groups was essential due to differences in roles, knowledge, and experience.

## 2.7. Quantitative data collection

A structured checklist-style questionnaire was administered to health personnel involved in E&CS implementation in Río Florido, including health personnel from DSVII and CSLL (see supplementary material). The questionnaire assessed four CFIF constructs: Content Details (10 questions), Frequency (5 questions), Duration (3 questions), and Coverage (15 questions). Adherence ≥80% was considered high implementation fidelity. The instrument was adapted from a questionnaire developed by Chipukuma et al. (2018) [16].

## 2.8. Qualitative analysis

Thematic analysis was conducted from semi-structured interviews with health personnel and community members. For health personnel, a priori categories based on CFIR (intervention, individual characteristics, implementation process, and internal/external context) were applied, followed by identification of emerging categories through open and axial coding. For community members, thematic categories were identified from recurrent patterns in testimonies. This approach allowed theoretical constructs to be contextualized in local practice and identified factors that hindered or facilitated effective implementation. Data organization and coding were performed using Microsoft Word 2021.

Additionally, the frequencies of barriers and facilitators reported by health personnel were compared between groups (DSVII and CSLL) using Fisher's exact test. Subsequently, a chi-square test was applied to determine whether certain barriers and/or facilitators occurred more or less frequently. All analyses were performed using R Studio ("Ghost Orchid" Release, RStudio Team 2021) [17].

## 2.9. Quantitative analysis

Due to the small sample size and high adherence rates, formal statistical comparisons are not feasible. To evaluate implementation fidelity, an instrument measuring four CFIF constructs was used: Content Details (maximum 30 points), Frequency (16 points), Duration (8 points), and Coverage (32 points), totaling 86 points as the maximum fidelity score. Individual adherence scores were calculated by dividing each participant's total by the maximum possible score. Means, standard deviations, and fidelity percentages were estimated for each construct and overall, both individually and by group (DSVII and CSLL).

Descriptive analyses were performed in Microsoft Excel. Means of fidelity were compared between the two groups using Student's t-tests for independent samples with equal variances (Ghost Orchid Release, RStudio Team 2021), to identify potential differences in program implementation.

## 2.10. Ethical considerations

The study was approved by the Research Ethics Committee of the National School of Public Health, University of Antioquia (approvals 21030002-00181-2022 and 21030002-00227-2022), by the Epidemiology Program Committee of the University of Antioquia (folio 21040004-MEPI-178–2022), and received institutional approval from DSVII (folio DSVII/SP/PS/N° 157–2022) and CSLL (folio DSVII/SP/PS/N° 144–2022) in Tapachula. Community permission was obtained from the President of the governing board of the ejido Río Florido. All participants provided written informed consent. Confidentiality, anonymity, and voluntary participation were ensured.

# 3. Results

## 3.1. Participant characteristics

In the Río Florido community, 96 residents were interviewed, of whom 60.4% had lived in the locality for more than 10 years. Among respondents, 73.9% were women, with a mean age of 41.8 years, while men had a mean age of 50.1 years.

Within the Health Secretariat, the two components showed similarities with some minor differences. In DSVII and CSLL, between 90% and 100% of health personnel held a university degree. Mean age ranged from 41 to 46 years. Female personnel were less represented in DSVII (44%) compared to CSLL (60%). All personnel had more than two years working for DSVII and CSLL. In DSVII, 83.3% of staff had been linked to the E&CS Program during this period, while all staff at CSLL had been affiliated. Regarding motivation for working at the Health Secretariat, 83.3% of DSVII personnel reported being recommended for their position, whereas all CSLL employees were recommended (Table 1).

## 3.2. Qualitative results

### 3.2.1. Barriers and facilitators perceived by health personnel.
Results were organized according to the a priori categories defined by the CFIR: Complexity, Design and quality of delivery, and Strength and quality of evidence.

**Table 1. Sociodemographic characteristics of health personnel working at the Tapachula Health Secretariat (Chiapas, Mexico) and linked to the E&CS Program in Río Florido.**

| Characteristic | DSVII | CSLL |
| --- | --- | --- |
| n | 18 | 5 |
| **Age** | | |
| Mean (±SD) | 46 (±4.4) | 41 (±7.04) |
| Median (IQR) | 48 (42–57) | 41 (39–49) |
| Minimum | 31 | 31 |
| Maximum | 72 | 52 |
| **Sex** | | |
| Female (%) | 8 (44.4) | 3 (60) |
| Male (%) | 10 (55.6) | 2 (40) |
| **Education level** | | |
| Secondary (%) | 2 (11.1) | 0 (0) |
| University (%) | 16 (88.9) | 5 (100) |
| **Time working at the Health Secretariat** | | |
| More than 2 years (%) | 18 (100) | 5 (100) |
| **Time working in E&CS** | | |
| Less than 6 months (%) | 1 (5.6)* | 0 (0) |
| 6–12 months (%) | 1 (5.6) | 0 (0) |
| 1–2 years (%) | 1 (5.6) | 0 (0) |
| More than 2 years (%) | 15 (83.3) | 5 (100) |
| **Reason for working at the Health Secretariat** | | |
| Recommended by the Health Secretariat (%) | 15 (83.3) | 5 (100) |
| Interest in dengue activities (%) | 1 (5.6) | 0 (0) |
| Other reasons (%) | 2 (11.1) | 0 (0) |
| **Reason for working in E&CS** | | |
| Recommended by Health District VII (%) | 18 (100) | 5 (100) |
| Interest in dengue activities (%) | 0 (0) | 0 (0) |
| Other reasons (%) | 0 (0) | 0 (0) |

SD: Standard deviation; IQR: Interquartile range.

*This individual had been working in the Tapachula Health Secretariat and in the E&CS program for more than 2 years in another district. Although transferred to DSVII only 6 months prior, they were considered eligible for the study due to participation in E&CS activities during the study period.

*Complexity:* Implementation was perceived as complex due to limited staff, resources, and supplies, which hindered activity execution. As one member of the health personnel noted, "These activities are complicated because we have very little staff and few supplies" (CSLL003). Community participation was also limited by the absence of social or economic incentives, affecting motivation to engage "If there is no social or monetary benefit for the community, people do not support the program" (CSLL002). A lack of interest and commitment from decision-makers, who prioritized other issues, was also reported: "There is apathy, a lack of interest… for them (decision-makers), there are other aspects they consider more important" (DSVII018). Logistical limitations, such as insufficient transport and administrative materials, further hindered daily operations, as reflected in the comment: "The Health Secretariat should show more interest in transportation and in providing photocopies and paperwork when we need them" (DSVII014). Finally, deficiencies in information dissemination to the community were identified, potentially influencing effective participation.

*Design and quality of delivery*: Personnel valued the educational materials used in the campaigns positively, perceiving them as appropriate, contextualized, and useful for conveying key messages to the community. One participant explained: "We use brochures, banners, digital information, posters… they are good-quality materials, and we always choose the ones that best fit the community's context" (DSVII003). This reflects not only the availability of resources but also the adaptation of content to the sociocultural characteristics of the setting, enhancing acceptability and comprehension.

*Strength and quality of evidence*: Personnel demonstrated solid knowledge of the program and its objectives, reinforcing confidence in the intervention strategy. As one participant described: "The campaign focuses on community certification, self-care, and healthy lifestyles. We talk about preventing diseases like dengue, so people learn to take care of themselves, with staff only acting as allies" (DSVII004). Emphasis was placed on promoting self-care, dengue prevention, and building healthy communities, aligning with the program goals.

Two emergent categories further improved understanding of the implementation process:

• Implementation challenges: operational and structural obstacles, such as lack of resources, technical complexity, resistance to change, and limited intersectoral coordination, which hindered achieving targets.

• Health personnel commitment: active involvement, willingness to work with the community, and trust-building networks, key factors for continuity of activities under adverse conditions.

The perceptions of the two health groups (DSVII and CSLL) did not show significant differences (Fisher's exact test, $p > 0.05$), so the data from both groups were combined. Health personnel (n = 23; 142 responses) identified various barriers and facilitators in the implementation of the E&CS program, with no significant differences detected among them, indicating a balanced pattern in their perceptions (Fig 2; $\chi^2 = 9.73$, df = 6, p = 0.136). The barriers included: 1) complex activities, 2) economic limitations, 3) lack of logistical support, and 4) lack of community support. The facilitators were: 1) use of contextually appropriate communication methods, 2) high commitment to community work, and 3) extensive knowledge and experience in health promotion and dengue prevention.

**3.2.2. Barriers and facilitators of implementation perceived by Río Florido residents.** Analysis of community residents' testimonies revealed diverse perceptions of the E&CS program and the work of health personnel. Three main categories were identified.

*Perception of program activities and health personnel visits*: The community recognized that health personnel carry out educational activities, yard cleaning, and larvicide distribution. However, in recent months, these visits have become irregular, generating distrust and a perception that prevention relies mainly on the population. For example, one resident stated:

"What they have to do is visit the community more frequently; otherwise, people do not want to follow instructions because there is no one keeping track" (ERF071).

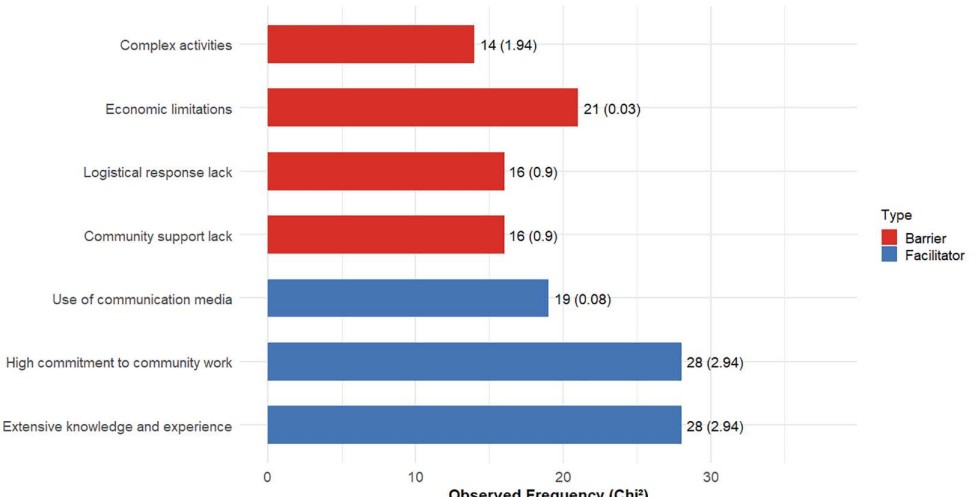

**Fig 2. Distribution of perceived barriers and facilitators among health personnel.** First number: observed frequencies; number in parenthesis: chi-square contributions.

Another commented: "Yes, we carried out the activities, they used to come and hold meetings at the ejido house, gave us talks about the mosquito, what we had to do to eliminate it and not get sick... but now they do not come anymore" (ERF021).

This irregularity affects the perception of program continuity and effectiveness.

*Relationship with health personnel*: Testimonies highlighted the friendliness and cordiality of the health personnel, as well as clarity in communication, elements that facilitate community collaboration:

"They are kind people and try to raise awareness about the importance of prevention"

(ERF045), and "The instructions are understandable and clear, their treatment towards us is very cordial" (ERF074).

This positive relationship promotes a favorable environment for prevention and community well-being.

*Knowledge and practices for dengue prevention*: The community demonstrated good knowledge and practice of preventive measures, such as eliminating standing water and disposing of containers, and recognized the seriousness of dengue: "Well, taking care of the water, removing standing water throughout the house, throwing away containers, being alert to the truck" (ERF018); "Yes, it is serious (dengue) and there have been cases, it is one of the diseases that has most affected the community" (ERF055); and "They have told us, and it depends on us not to let water stagnate, and we try to keep the house clean, for example, in the morning I cleaned the yard, it stayed very clean" (ERF053).

In the community of Río Florido (n = 96; 821 responses), various barriers and facilitators perceived by residents were identified. Statistically significant differences were detected among the frequencies of these perceptions (Fig 3; $\chi^2 = 83.16$, df = 11, $p < 0.001$). The perceptions that differed the most from all others were the barriers "Insufficient attention from the health center" ($\chi^2 = 25.07$) and "SSA perceived as responsible for dengue" ($\chi^2 = 21.57$). These were also the least frequently mentioned barriers overall, while "Lack of consistency in visits" was the most frequently mentioned ($\chi^2 = 5.05$). Among the facilitators, no single one stood out. The most frequently mentioned were "Recognition of the E&CS program," "Friendly and respectful relationship with health personnel," "Belief in the success of the program," "Elimination of breeding sites," and "Knowledge about dengue prevention".

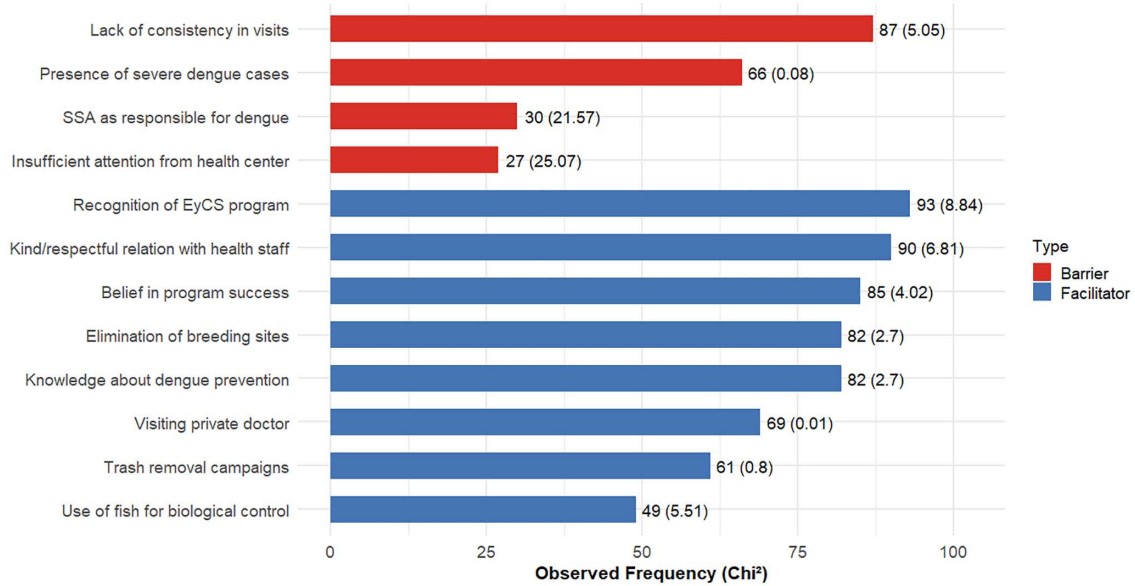

**Fig 3. Distribution of perceived barriers and facilitators among community members in Río Florido.** First number: observed frequencies; number in parenthesis: chi-square contributions.

### 3.3. Quantitative results

**3.3.1. Implementation fidelity.** The percentage adherence to the E&CS program constructs Content Details, Frequency, Duration and Coverage and the overall adherence was calculated for health personnel from DSVII (n = 18) and CSLL (n = 5). No statistically significant differences were found in any of the evaluated constructs or in the overall adherence, which ranged between 96% and 98%, indicating homogeneous and consistent adherence across both groups (Fig 4).

Content details adherence ranged from 67% to 100%, with a mean of 92% ± 7.62%. Monthly supervision was reported between 80% and 100%, feedback from supervisors was positively valued by over 90% of personnel, and approximately 80–85% reported having received training or updates on E&CS about two years ago. All personnel understood the objectives of the program, and between 90% and 100% confirmed the availability of inventories for the campaign.

Frequency adherence was very high, ranging from 75% to 100%, with all personnel submitting five or more reports in the last six months. Monthly sensitization talks on dengue prevention were conducted, and the community was informed about breeding site searches and source reduction campaigns. The mean adherence for frequency was 98% ± 7.61%.

Duration adherence ranged from 38% to 100%, reflecting the frequency of household visits, with 41–50 homes inspected monthly in Río Florido, covering 100% of households. The mean adherence for duration was 97% ± 16.54%. Coverage ranged from 72% to 100%, with nearly complete dengue case follow-up and reporting, and almost all personnel providing information to the community on preventive measures, symptom identification, and proper use of mosquito nets. The mean adherence for coverage was 98% ± 9.95%.

Overall, these results demonstrate high fidelity of implementation across all evaluated constructs, consistent with the objectives of the E&CS program. Due to the small sample sizes and high adherence rates, formal statistical testing was not appropriate. Tables 2 and 3 provide the detailed adherence scores for each construct and overall program adherence for individual personnel.

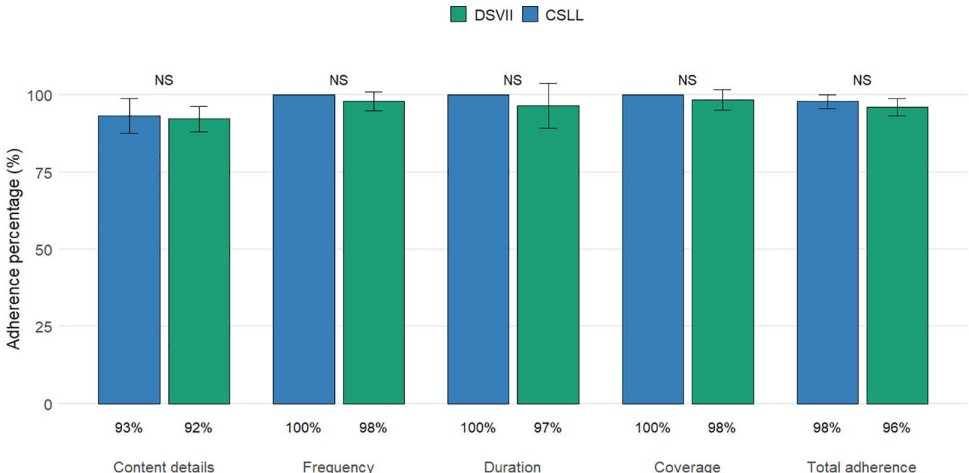

**Fig 4. Adherence (%) to E&CS program constructs by health personnel group (DSVII and CSLL).** Student's t-test; NS: p > 0.05.

## 4. Discussion

The findings of this study indicate that the E&CS program is implemented with high fidelity by health personnel in the community of Río Florido, Tapachula. However, specific gaps were identified, such as in the "Content Details" component, and a community perception of irregular visits, which contrasts with the reports of the interviewed health personnel. This discrepancy highlights a gap between the technical execution of the program and the lived experience of the community. Differences between CSLL and DSVII within this component are related to aspects such as supervision frequency, timing and update of training, and availability of supplies. These gaps reflect limited articulation between both components, likely conditioned by logistical and budgetary constraints. This lack of coordination may translate into inconsistent actions, implementation gaps, or variations in how the program is executed, affecting the community's perception of program continuity and institutional commitment to dengue control.

These results are consistent with previous studies in the Mesoamerican region that document common challenges in implementing community dengue programs, such as low citizen participation, lack of intersectoral coordination, and poor sustainability of strategies [18]. Additionally, studies such as Saré et al. [8] show that even with robust conceptual frameworks, failures in community mobilization and coordination with the health sector affect implementation fidelity and expected outcomes [9].

The finding that adequate community knowledge about dengue prevention measures does not always translate into effective practices aligns with previous research in southeastern Mexico [18]. This reinforces the idea that interventions must go beyond health education, incorporating approaches that address structural and motivational factors. Fidelity in implementing the E&CS strategy alone does not guarantee the expected impact if it is not accompanied by community ownership and contextual adaptability.

From a public health perspective, irregular visits may negatively affect early case detection, continuous education, and breeding site elimination, compromising program sustainability [19]. However, it is necessary to emphasize that dengue is a disease determined by environmental, socioeconomic, and local governance conditions [20]. Therefore, focusing exclusively on visit frequency may lead to a reductionist interpretation of the problem. An integrated approach considering these structural determinants is required to strengthen the implementation and impact of community interventions.

**Table 2. Adherence percentages (item scores) by constructs (Content Details, Frequency, Duration, and Coverage) and overall program adherence among E&CS personnel from the DSVII.**

| Participant | Content Details (30) | Frequency (16) | Duration (8) | Coverage (32) | Overall Adherence (86) |
|---|---|---|---|---|---|
| 1 | 67% (20) | 100% (16) | 100% (8) | 100% (32) | 88% (76) |
| 2 | 83% (25) | 100% (16) | 100% (8) | 100% (32) | 94% (81) |
| 3 | 87% (26) | 94% (15) | 100% (8) | 100% (32) | 94% (81) |
| 4 | 97% (29) | 100% (16) | 100% (8) | 100% (32) | 99% (85) |
| 5 | 83% (25) | 100% (16) | 100% (8) | 100% (32) | 94% (81) |
| 6 | 90% (27) | 94% (15) | 100% (8) | 100% (32) | 95% (82) |
| 7 | 100% (30) | 100% (16) | 100% (8) | 100% (32) | 100% (86) |
| 8 | 97% (29) | 100% (16) | 100% (8) | 100% (32) | 99% (85) |
| 9 | 93% (28) | 100% (16) | 100% (8) | 100% (32) | 98% (84) |
| 10 | 93% (28) | 100% (16) | 100% (8) | 100% (32) | 98% (84) |
| 11 | 90% (27) | 100% (16) | 100% (8) | 97% (31) | 95% (82) |
| 12 | 93% (28) | 100% (16) | 100% (8) | 100% (32) | 98% (84) |
| 13 | 100% (30) | 100% (16) | 100% (8) | 100% (32) | 100% (86) |
| 14 | 100% (30) | 100% (16) | 100% (8) | 100% (32) | 100% (86) |
| 15 | 100% (30) | 100% (16) | 100% (8) | 100% (32) | 100% (86) |
| 16 | 100% (30) | 100% (16) | 100% (8) | 100% (32) | 100% (86) |
| 17 | 93% (28) | 75% (12) | 38% (3) | 72% (23) | 77% (66) |
| 18 | 93% (28) | 100% (16) | 100% (8) | 100% (32) | 98% (84) |
| Mean±SD (Mean score) | 92%±7.62 (27.7) | 98%±7.61 (15.7) | 97%±16.54 (7.7) | 98%±9.95 (31.4) | 96%±5.36 (82.9) |

**Table 3. Adherence percentages (item scores) by constructs (Content Details, Frequency, Duration, and Coverage) and overall program adherence among E&CS personnel from the CSLL.**

| Participant | Content Details (30) | Frequency (16) | Duration (8) | Coverage (32) | Overall Adherence (86) |
|---|---|---|---|---|---|
| 1 | 93% (28) | 100% (16) | 100% (8) | 100% (32) | 98% (84) |
| 2 | 93% (28) | 100% (16) | 100% (8) | 100% (32) | 98% (84) |
| 3 | 93% (28) | 100% (16) | 100% (8) | 100% (32) | 98% (84) |
| 4 | 100% (30) | 100% (16) | 100% (8) | 100% (32) | 100% (86) |
| 5 | 87% (26) | 100% (16) | 100% (8) | 100% (32) | 95% (82) |
| Mean±SD (Mean score) | 93%±4.6 (28.0) | 100% (16) | 100% (8) | 100% (32) | 98%±1.79 (84.0) |

Symbolic interactionism theory, developed by George H. Mead and formalized by Herbert Blumer, asserts that people act based on the meanings they assign to objects, people, and situations. These meanings emerge and transform through social interaction [21]; and this perspective is useful for analyzing how communities interpret the presence or absence of health personnel, which directly affects their willingness to participate in collective interventions.

For example, a study in Hanoi, Vietnam, found that low interaction with health personnel and the perception of weak communication between institutional and community actors generated distrust and reduced community participation in dengue vector control activities [7]. Similarly, a study in Ouagadougou, Burkina Faso, observed that the partial success of a community intervention was linked to how the community interpreted the legitimacy and commitment of external facilitators [8].

These empirical findings show that weak relationships between community actors and health personnel can generate distrust, demotivation, and low participation, undermining the principles of shared responsibility and empowerment that support community interventions.

Among the potential limitations of this study is the small number of health personnel interviewed and surveyed, which could have generated imprecision in estimates and introduced selection bias. However, all DSVII health personnel responsible for municipal-level program activities, as well as all CSLL personnel in charge of case notification and direct community contact, were included. Therefore, although the sample size is small, it represents the teams responsible for implementation, allowing a representative approximation of their experiences regarding program fidelity. In the case of the community, the sample calculation and selection strategy ensured representativeness of participants, given the group's homogeneity, minimizing the risk of bias.

Additionally, there is a possibility that some responses, from both health personnel and community participants, may not faithfully reflect their real perceptions or experiences, due to factors such as social desirability or the perception of being evaluated. This potential bias was recognized and addressed through a clear presentation of the study, assurance of confidentiality, and obtaining informed consent to create an environment of trust that promoted honest responses.

These results suggest the need to strengthen the participatory components of the program, incorporate community feedback systematically, and ensure coordination across operational levels. The use of implementation research frameworks, such as CFIF and CFIR, could provide a broader understanding of implementation processes and contextual barriers limiting program effectiveness [22]. The use of quantitative and qualitative data also contributes to a better understanding of this issue [23].

Fidelity in implementing the E&CS program in Río Florido was high, although significant limitations in coverage and frequency were identified, mainly attributable to logistical and institutional barriers. The effectiveness of interventions like E&CS depends not only on technical design but also on adaptation to the local context, operational capacity, and active community participation.

To strengthen programming fidelity and impact, it is necessary to:

- Ensure minimum sustainable resources (materials, transportation, personnel)

- Improve coordination between SSA operational levels

- Strengthen community awareness processes from an intercultural perspective

This study contributes to understanding the factors affecting the real implementation of community interventions for dengue control and provides useful evidence for decision-makers seeking to improve public health in marginalized rural contexts.

## Acknowledgments

We sincerely thank the authorities of the SSA of Tapachula, the staff of DSVII and CSLL, as well as the community of the ejido Río Florido for their generous collaboration. This study would not have been possible without the willingness, trust, and active participation of the residents and health personnel. We also appreciate the academic and methodological support provided by the professors and advisors from the University of Antioquia and El Colegio de la Frontera Sur (ECOSUR). Furthermore, we express our special recognition to Javier Francisco Valle Mora, engineer at the Tecnológico Nacional de México (TECNM) Campus Tapachula, for his guidance in the statistical analysis of the results of this work.

## Author contributions

**Conceptualization:** Miguel Mazariego-Longoria, Carlos Rojas-Arbeláez, Abel Jiménez-Alejo, Ariane Dor, Guillermo Rúa-Uribe.

**Data curation:** Miguel Mazariego-Longoria, Héctor Vélez-Santamaría, Ariane Dor.

**Formal analysis:** Miguel Mazariego-Longoria, Ariane Dor, Guillermo Rúa-Uribe.

**Funding acquisition:** Carlos Rojas-Arbeláez.

**Investigation:** Miguel Mazariego-Longoria, Héctor Vélez-Santamaría, Abel Jiménez-Alejo, Ariane Dor, Guillermo Rúa-Uribe.

**Methodology:** Miguel Mazariego-Longoria, Carlos Rojas-Arbeláez, Héctor Vélez-Santamaría, Abel Jiménez-Alejo, Ariane Dor, Guillermo Rúa-Uribe.

**Project administration:** Miguel Mazariego-Longoria, Carlos Rojas-Arbeláez.

**Supervision:** Carlos Rojas-Arbeláez, Héctor Vélez-Santamaría, Abel Jiménez-Alejo, Ariane Dor, Guillermo Rúa-Uribe.

**Validation:** Miguel Mazariego-Longoria, Abel Jiménez-Alejo, Ariane Dor, Guillermo Rúa-Uribe.

**Visualization:** Miguel Mazariego-Longoria.

**Writing – original draft:** Miguel Mazariego-Longoria.

**Writing – review & editing:** Carlos Rojas-Arbeláez, Héctor Vélez-Santamaría, Abel Jiménez-Alejo, Ariane Dor, Guillermo Rúa-Uribe.

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
