## [Decision Letter · Decision Letter 0]

25 Nov 2025

Dear Dr. Rojas,

Thank you for submitting your manuscript to PLOS ONE. After careful consideration, we feel that it has merit but does not fully meet PLOS ONE’s publication criteria as it currently stands. Therefore, we invite you to submit a revised version of the manuscript that addresses the points raised during the review process.

We look forward to receiving your revised manuscript.

Kind regards,

José Ramos-Castañeda, M.Sc., Ph.D

Academic Editor

PLOS ONE

Journal Requirements:

https://journals.plos.org/plosone/s/file?id=ba62/PLOSOne_formatting_sample_title_authors_affiliations.pdf....

2. Please include a complete copy of PLOS’ questionnaire on inclusivity in global research in your revised manuscript. Our policy for research in this area aims to improve transparency in the reporting of research performed outside of researchers’ own country or community. The policy applies to researchers who have travelled to a different country to conduct research, research with Indigenous populations or their lands, and research on cultural artefacts. The questionnaire can also be requested at the journal’s discretion for any other submissions, even if these conditions are not met. Please find more information on the policy and a link to download a blank copy of the questionnaire here: https://journals.plos.org/plosone/s/best-practices-in-research-reporting. Please upload a completed version of your questionnaire as Supporting Information when you resubmit your manuscript

3, Thank you for stating the following financial disclosure:

“Funded by TDR, the Special Programme for Research and Training in Tropical Diseases, hosted at the World Health Organization and co-sponsored by UNICEF, UNDP, the World Bank, and WHO (TDR grant number B40323). http://www.who.int/tdr/about/en/.”

4. Please be informed that funding information should not appear in the Acknowledgments section or other areas of your manuscript. We will only publish funding information present in the Funding Statement section of the online submission form. Please remove any funding-related text from the manuscript.

5. Thank you for providing your underlying data as Supporting Information.

We note that the data set contains text or data that is not in English. Please note that PLOS is an English-language publisher, so we require data sets to be provided in English as well. Please upload an English-language version of your data set.

This will also allow us to determine if your data follows PLOS standards per our Data Availability policy here: https://journals.plos.org/plosone/s/data-availability....

6. We note that Figure 1 in your submission contain [map/satellite] images which may be copyrighted. All PLOS content is published under the Creative Commons Attribution License (CC BY 4.0), which means that the manuscript, images, and Supporting Information files will be freely available online, and any third party is permitted to access, download, copy, distribute, and use these materials in any way, even commercially, with proper attribution. For these reasons, we cannot publish previously copyrighted maps or satellite images created using proprietary data, such as Google software (Google Maps, Street View, and Earth). For more information, see our copyright guidelines: http://journals.plos.org/plosone/s/licenses-and-copyright.

a. You may seek permission from the original copyright holder of Figure(s) [#] to publish the content specifically under the CC BY 4.0 license.

Reviewers' comments:

Reviewer's Responses to Questions

**Comments to the Author**

1. Is the manuscript technically sound, and do the data support the conclusions?

Reviewer #1: Yes

Reviewer #2: No

Reviewer #3: Yes

2. Has the statistical analysis been performed appropriately and rigorously?

Reviewer #1: Yes

Reviewer #2: No

Reviewer #3: Yes

3. Have the authors made all data underlying the findings in their manuscript fully available?

Reviewer #1: Yes

Reviewer #2: Yes

Reviewer #3: Yes

4. Is the manuscript presented in an intelligible fashion and written in standard English?

Reviewer #1: Yes

Reviewer #2: Yes

Reviewer #3: Yes

Reviewer #1: The presented manuscript offers an important approach to a prevalent public health problem worldwide. I believe that the findings identified in the qualitative approach, which enriches the study, should be forther ditail.

Reviewer #2: Dengue prevention in Mexico relies on vector control, with a particular focus on eliminating Aedes aegypti breeding sites. How much of dengue control activities are directed against breeding sites? The problem with initiatives directed against breeding sites is to evaluate number, type, size, capacity frequency of positiveness and productivity so as to direct specific efforts to what containers /breeding sites are most productive and when interventions should be implemented. While authors mentioned that a pilot study was performed no data on the data explore is presented. They assume ha the reality in Rio Florido is not different from any other place in the city or the state.

Evaluate the implementation fidelity of the E&CS program in Río Florido promotes community practices to eliminate Aedes aegypti breeding sites. Institutionally driven intervention that aims to foster social participation in health promotion and disease prevention through the transformation of physical and social environments is a very complex task. The authors do not mention any particular trait of the entomological situation in Rio Florido that could influence or enrich the implementation of E&CS program.

As stated by the authors protocol includes: 1) participatory diagnosis of health risks and environmental hazards; 2) joint identification of intervention priorities; 3) development of a community action plan; 4) execution of activities such as cleanup campaigns, home visits, and educational workshops; and 5) monitoring and evaluation of results with feedback to the community. These stages are facilitated by health personnel in collaboration with community leaders, health committees, and other social actors, with the goal of fostering sustainable practices that reduce vector presence and improve environmental conditions. Without entomological situational diagnosis it is doom to fail in terms of participation and adherence.

To evaluate the degree to which an intervention is delivered as intended, -Implementation fidelity-, is a critical component to ensure the expected impact. The evaluation of Implementation fidelity is understood as a complex enterprise by the health personnel and the community.

In this study, they intend to evaluate the implementation fidelity of the Health Secretariat personnel to key E&CS program components and the perceived barriers and facilitators from institutional and community perspectives through the Consolidated Framework for Implementation Research (CFIR) , which systematically examines internal and external contextual factors affecting implementation (qualitative data), and the Conceptual Framework for Implementation Fidelity (CFIF) (6), which considers dimensions such as adherence to Content Details, Duration, Frequency, and Coverage (quantitative data). The authors fail to illustrate the dimensions or components of content, duration, frequency and coverage of the program in a clear manner. Results are very simplistic, poorly described and figures are not clear in terms of presenting frequencies or %, chi square values or significance. Approach is descriptive and no analysis is presented.

Reviewer #3: Dear Authors,

My acknowledgement for your manuscript about fidelity implementation of a dengue program. I found your analysis clear and your methodology adequate. Aiming to improve your presentation, I have some suggestion for you:

1) It is necesary to provide the objective of the study or the objerctive of the evaluation. Generally, it is provided in the last paragraph of the introduction.

2) I found also relevant to describe the objectives of the analyzed program, for better understand the barriers described in the manuscript.

3) It is not clear why to highlight in red some lines.

With kind regards,

.

Reviewer #1: No

Reviewer #2: No

Reviewer #3: **Yes:** Emanuel Orozco Núñez, Ma.Emanuel Orozco Núñez, Ma.Emanuel Orozco Núñez, Ma.Emanuel Orozco Núñez, Ma.

---

## [Author Response · Author response to Decision Letter 1]

8 Dec 2025

Dear Editor and Reviewers,

We sincerely thank you for your careful review of our manuscript “Implementation fidelity of a community-based Aedes aegypti breeding site elimination program for dengue control in southeastern Mexico.” We greatly appreciate the constructive comments and suggestions provided, which have helped us improve the clarity, rigor, and overall quality of the manuscript.

In the revised manuscript, we have addressed all reviewer comments, incorporated requested clarifications, and provided additional supporting information as required by the journal. As noted below, our point-by-point responses indicate the corresponding line numbers from the clean version of the manuscript (i.e., without track changes).

Journal Requirements:

Response: Done. The manuscript was revised to fully comply with PLOS ONE style guidelines and file naming conventions.

Response: Done. The completed PLOS inclusivity questionnaire was added as Supporting Information in the revised manuscript.

“Funded by TDR, the Special Programme for Research and Training in Tropical Diseases, hosted at the World Health Organization and co-sponsored by UNICEF, UNDP, the World Bank, and WHO (TDR grant number B40323). http://www.who.int/tdr/about/en/.”

Response: Done. The funders had no role in study design, data collection and analysis, decision to publish, or preparation of the manuscript.

4. Please be informed that funding information should not appear in the Acknowledgments section or other areas of your manuscript. We will only publish funding information present in the Funding Statement section of the online submission form. Please remove any funding-related text from the manuscript.

Response: Done. All funding information was removed from the Acknowledgments section and other manuscript sections, remaining only in the Funding Statement section of the online submission form.

5. Thank you for providing your underlying data as Supporting Information.

We note that the data set contains text or data that is not in English. Please note that PLOS is an English-language publisher, so we require data sets to be provided in English as well. Please upload an English-language version of your data set.

This will also allow us to determine if your data follows PLOS standards per our Data Availability policy here: https://journals.plos.org/plosone/s/data-availability.

Response: Done. An English-language version of the underlying data set was prepared and uploaded as Supporting Information.

6. We note that Figure 1 in your submission contains [map/satellite] images which may be copyrighted. All PLOS content is published under the Creative Commons Attribution License (CC BY 4.0), which means that the manuscript, images, and Supporting Information files will be freely available online, and any third party is permitted to access, download, copy, distribute, and use these materials in any way, even commercially, with proper attribution. For these reasons, we cannot publish previously copyrighted maps or satellite images created using proprietary data, such as Google software (Google Maps, Street View, and Earth). For more information, see our copyright guidelines: http://journals.plos.org/plosone/s/licenses-and-copyright.

a. You may seek permission from the original copyright holder of Figure(s) [#] to publish the content specifically under the CC BY 4.0 license.

USGS EROS (public domain): http://eros.usgs.gov/#

Response: Done. Figure 1 was created entirely by the author MAML in QGIS using open-access geospatial data from INEGI, which are publicly available and licensed under CC BY 4.0. The figure caption has been updated to explicitly indicate that it was created with open-access data compatible with the CC BY 4.0 license. No copyrighted material remains in the figure.

Response: N/A

Reviewer #1 observations:

The presented manuscript offers an important approach to a prevalent public health problem worldwide. I believe that the findings identified in the qualitative approach, which enriches the study, should be forther ditail.

Response:

We appreciate the reviewer’s valuable comment. In response, we have incorporated illustrative participant quotes directly into the qualitative section to more clearly show the foundations of the identified categories. We believe this addition strengthens the transparency of the analysis while keeping the section within the journal’s length and structural requirements.

Reviewer #2 observations:

Dengue prevention in Mexico relies on vector control, with a particular focus on eliminating Aedes aegypti breeding sites. How much of dengue control activities are directed against breeding sites? The problem with initiatives directed against breeding sites is to evaluate number, type, size, capacity frequency of positiveness and productivity so as to direct specific efforts to what containers /breeding sites are most productive and when interventions should be implemented.

Response: We thank the reviewer for his/her observations, to respond to the question “How much of dengue control activities are directed against breeding sites?” It is important to clarify that dengue control in Mexico involves several institutional areas within the health system. In general, the main areas participating in dengue-related activities are Entomovirology, Epidemiological Surveillance, Clinical Management of Dengue, Primary Care or Community Health, Vector Control, and Health Promotion, within which the E&CS program operates.

Within this structure, two areas focus specifically on activities related to Aedes aegypti breeding sites. The Vector Control area is responsible for technical measurements and detailed monitoring of breeding sites, including entomological surveillance. In turn, E&CS dedicates an important part of its work to the elimination of visible breeding sites through the identification of risk containers, basic sanitation, removal of accessible water-holding containers, and the promotion of household practices to prevent water accumulation.

For this reason, our study focused exclusively on evaluating fidelity to the activities formally established by E&CS, which are centered on intervening the most common breeding sites in the community.

While authors mentioned that a pilot study was performed no data on the data explore is presented. They assume ha the reality in Rio Florido is not different from any other place in the city or the state.

Response: We thank the reviewer for his/her observations and we modified the Material and Methods: 2.4 Pilot testing section (Line 128-145):

“A pilot study was conducted to pre-evaluate the data collection instruments under real field conditions and to verify whether the selected variables were adequate in terms of clarity, application time, and participant comprehensibility. Semi-structured interviews (qualitative) and structured questionnaires (quantitative) were administered to health personnel (n = 4) from the Health Promotion area of the health center located in Segunda Sección de Tinajas, municipality of Tapachula, Chiapas. Segunda Sección de Tinajas is a rural locality situated approximately between 36 and 44 meters above sea level and is part of the coastal plain of the Soconusco region; it has an estimated population of 757 inhabitants and high coverage of basic services such as piped water, electricity, and sanitation (13).

Semi-structured interviews were also conducted with residents of the ejido Francisco I. Madero (n = 10), selected due to its sociodemographic similarity to Río Florido, the main study site. This ejido is in the municipality of Tapachula at approximately 29 meters above sea level and has an estimated population of 678 inhabitants; the available basic housing and service indicators describe demographic and dwelling characteristics typical of rural communities in the coastal plain of the Soconusco region (14).”

Evaluate the implementation fidelity of the E&CS program in Río Florido promotes community practices to eliminate Aedes aegypti breeding sites. Institutionally driven intervention that aims to foster social participation in health promotion and disease prevention through the transformation of physical and social environments is a very complex task. The authors do not mention any particular trait of the entomological situation in Rio Florido that could influence or enrich the implementation of E&CS program.

Response: We thank the reviewer for his/her observations and we added a paragraph in the Introduction section (Line 54-63):

“Importantly, entomological evidence from rural localities in southern Chiapas, including Río Florido, shows that Aedes aegypti and Aedes albopictus coexist in the area, with marked seasonal fluctuations and persistent egg abundance throughout the year. Baseline data from longitudinal oviposition monitoring in Río Florido demonstrate higher Aedes activity during the rainy season and stable presence during the dry season, reflecting continuous vector pressure in the community (4). These ecological conditions highlight the relevance of targeting breeding-site elimination and community engagement strategies, as the local entomological context may influence both the implementation requirements and the fidelity of programs such as E&CS.”

The cited article was added to sustain this fact (Line 519):

“4. Marina CF, Bond JG, Hernández-Arriaga K, Valle J, Ulloa A, Fernández-Salas I, Carvalho DO, Bourtzis K, Dor A, Williams T, et al. Population dynamics of Aedes aegypti and Aedes albopictus in two rural villages in southern Mexico: baseline data for an evaluation of the sterile insect technique. Insects. 2021;12(1):58. doi:10.3390/insects12010058”

The addition of this article in the bibliography changed the sequences of all the subsequent cited articles.

As stated by the authors protocol includes: 1) participatory diagnosis of health risks and environmental hazards; 2) joint identification of intervention priorities; 3) development of a community action plan; 4) execution of activities such as cleanup campaigns, home visits, and educational workshops; and 5) monitoring and evaluation of results with feedback to the community. These stages are facilitated by health personnel in collaboration with community leaders, health committees, and other social actors, with the goal of fostering sustainable practices that reduce vector presence and improve environmental conditions. Without entomological situational diagnosis it is doom to fail in terms of participation and adherence.

To evaluate the degree to which an intervention is delivered as intended, -Implementation fidelity-, is a critical component to ensure the expected impact. The evaluation of Implementation fidelity is understood as a complex enterprise by the health personnel and the community.

Response: We thank the reviewer for his/her observations. Although entomological surveillance is a valuable component of comprehensive vector control, we respectfully disagree with the idea that, within the E&CS framework, the absence of a formal entomological situational diagnosis “dooms” participation and adherence. The E&CS programme is explicitly designed for communities with high social and health vulnerability, unfavourable environmental and social conditions, and clear gaps in access to services, where there is also a need and potential for active municipal and community participation. In these settings, community engagement is primarily driven by the recognition of everyday environmental and sanitary problems (water storage, waste, household containers) jointly identified with residents, rather than by formal entomological indices.

According to national guidelines, ECS is a health promotion and environmental management strategy and does not require an entomological situational diagnosis as an operational prerequisite. Our assessment of implementation fidelity was therefore aligned with what E&CS actually mandates and what health personnel are trained to deliver: community mobilization, participatory diagnosis of environmental and sanitary risks,

---

## [Decision Letter · Decision Letter 1]

3 Mar 2026

Implementation fidelity of a community-based Aedes aegypti breeding site elimination program for dengue control in southeastern Mexico

PONE-D-25-50635R1

Dear Dr. Rojas,

We’re pleased to inform you that your manuscript has been judged scientifically suitable for publication and will be formally accepted for publication once it meets all outstanding technical requirements.

Kind regards,

Miquel Vall-llosera Camps

Senior Staff Editor

PLOS One

Reviewers' comments:

Reviewer's Responses to Questions

**Comments to the Author**

Reviewer #3: All comments have been addressed

2. Is the manuscript technically sound, and do the data support the conclusions?

Reviewer #3: Yes

3. Has the statistical analysis been performed appropriately and rigorously?

Reviewer #3: Yes

4. Have the authors made all data underlying the findings in their manuscript fully available?

Reviewer #3: Yes

5. Is the manuscript presented in an intelligible fashion and written in standard English?

Reviewer #3: Yes

Reviewer #3: Thank you for considering my suggestions. This manuscript will be useful for improving local conditions, but also to improve dengue control in some other places.

With kind regards,

.

Reviewer #3: **Yes:** Emanuel Orozco Núñez, Ma.Emanuel Orozco Núñez, Ma.Emanuel Orozco Núñez, Ma.Emanuel Orozco Núñez, Ma.

---

## [Editor Report · Acceptance letter]

PONE-D-25-50635R1

PLOS One

Dear Dr. Rojas,

I'm pleased to inform you that your manuscript has been deemed suitable for publication in PLOS One. Congratulations! Your manuscript is now being handed over to our production team.

Kind regards,

on behalf of

Dr. Miquel Vall-llosera Camps

Staff Editor

PLOS One